# Yellow fever surveillance suggests zoonotic and anthroponotic emergent potential

Alisa Aliaga-Samanez [1 ✉], Raimundo Real [1,2], Marina Segura [3], Carlos Marfil-Daza[1] & Jesús Olivero [1,2]

Yellow fever is transmitted by mosquitoes among human and non-human primates. In the last decades, infections are occurring in areas that had been free from yellow fever for decades, probably as a consequence of the rapid spread of mosquito vectors, and of the virus evolutionary dynamic in which non-human primates are involved. This research is a patho-geographic assessment of where enzootic cycles, based on primate assemblages, could be amplifying the risk of yellow fever infections, in the context of spatial changes shown by the disease since the late 20th century. In South America, the most relevant spread of disease cases affects parts of the Amazon basin and a wide area of southern Brazil, where forest fragmentation could be activating enzootic cycles next to urban areas. In Africa, yellow fever transmission is apparently spreading from the west of the continent, and primates could be contributing to this in savannas around rainforests. Our results are useful for identifying new areas that should be prioritised for vaccination, and suggest the need of deep yellow fever surveillance in primates of South America and Africa.

[1] Grupo de Biogeografía, Diversidad y Conservación, Departamento de Biología Animal, Facultad de Ciencias, Universidad de Málaga, 29071 Malaga, Spain.
[2] Instituto IBYDA, Centro de Experimentación Grice-Hutchinson, Malaga, Spain. [3] Centro de Vacunación Internacional, Ministerio de Sanidad, Consumo y Bienestar Social, Estación Marítima, Recinto del Puerto, Muelle 3, 29001 Malaga, Spain. ✉email: alisaliaga@uma.es

Yellow fever, an acute viral haemorrhagic disease, was the first human pathology to be attributed to a virus, and the first demonstrated to be transmitted by arthropods[1,2]. This disease is caused by an arbovirus of the family *Flaviviridae*. This pathogen can be transmitted by mosquitoes of genus *Aedes* in America and Africa, and of genera *Aedes*, *Sabethes* and *Haemagogus* in America, through three transmission cycle types: the sylvatic cycle (between primates), the urban epidemic cycle (between humans), and the intermediate or savannah cycle in Africa (between humans that live near the jungle)[3]. The yellow fever mosquito (*Aedes aegypti*) and the Asian tiger mosquito (*A. albopictus*) represent a major risk for the spread of yellow fever, the former being the main vector in urban areas and the latter a bridge vector between the forest and the urban area[4]. The virus is principally maintained by the sylvatic cycle where it evolves[5], and where primate-human zoonotic transmissions through different mosquito species are also possible[6].

The disease has undergone many changes over the past few centuries. The yellow fever virus is native to Africa where it may have emerged around 3000 years ago[7]. The virus was transported by slave-trade ships in the 15th and the 16th centuries from Africa to the Americas[1]. The later geographical expansion during the 17th and the 18th centuries was closely linked to the spread of *A. aegypti* through the shipping industry and commerce[8]. The number of yellow fever cases decreased in the mid-1900s in Francophone Africa (e.g., most of north-west and central Africa) by vaccination[9], and in the Americas by effective controls on the principal urban vector, *A. aegypti*[10]. Since the late 20th century, however, there has been a resurgence of yellow fever in Africa and America[6]. At least 200,000 cases and 30,000 deaths were reported in 1990 worldwide[11]. In Africa, virulent outbreaks affected urban areas of Angola and the Democratic Republic of the Congo (DRC) between December 2015 and July 2016[12]. Shortly after, at the end of 2016, outbreaks started to occur in South Brazil, in areas that had been free from yellow fever cases for decades, probably following a southward path from Trinidad and Tobago[13]. Since then, new cases have been reported in Surinam, Nigeria, and French Guyana, where the last cases dated back to 1971, 1996, and 1998, respectively[14–16]. Consequently, despite the fact that control policies were able to virtually eliminate yellow fever in wide areas of the globe, the WHO insists in stating that prevention efforts should not be abandoned[17]. Instead, in order to prevent and respond efficiently to the occurrence of new outbreaks, risk areas should be delineated according to geographic and environmental factors related to yellow fever[6,18].

A suitable methodological and conceptual framework for achieving the comprehension and prediction of zoonotic outbreaks is "pathogeography"[19], from which the geographic distribution of zoonotic diseases is analyzed on the basis of a multilevel factor approach that considers ecological and human drivers, as well as the biogeography of animal species involving vectors, reservoirs and other roles in the zoonotic cycles[20,21]. The most recent model predicting the risk of infection by the yellow fever virus[6] contemplates all these factors, producing risk maps based on records of infection in humans from 1970 to 2016. The study suggests that the presence of *A. aegypti*, combined with the zoonotic potential for infections (given the presence of potential primate hosts), contributes the receptivity of yellow fever transmission in new regions such as Asia through the spread of the virus or through importation. So, if the current situation points to a yellow fever geographic spread, the forecasting of future trends will need pathogeographical analyses based on the spatiotemporal context.

Sylvatic cycles favour the existence of evolutionary dynamics that have recently led to new yellow fever virus lineages in South America[13]. These cycles, involving primates and a range of vector species[22], not only are the scenario of the virus diversification, but can also contribute to amplifying the risk of transmission to humans[23–25]. The globalization of transports, and the expansion of vectors that are themselves evolving, help new virus lineages spread and increase the risk of transmission to humans[5,26]. Once the new lineage has arrived in an area, it can enter in the local enzootic cycles and be exposed to selective pressures potentially leading to new viral variants[18]. There are precedents in the analysis of primate influence on infection rates. Hamrick et al.[18], in 2017, considering eight nonhuman primate genera in South America, calculated that the probability of yellow fever occurrence at the county level doubled with each additional genus. In 2018, Shearer et al.[6] carried out a study of the local speed at which human individuals are expected to be infected with the yellow fever virus, for which they included non-human primate distributions as variables in their models. That study was, thus, focused on the virus infection efficiency, whose geographic variation was analized within the area that is already suitable for the occurrence of yellow fever according to risk zones (47 countries of America and Africa) estimated by Jentes et al. in 2010[27]. Gaythorpe et al. in 2021[28], examined the impact of vaccination at the provincial level, taking into account the distribution of three non-human primate families as predictors in their models. Further studies are needed, however, to understand the extent and geography of the nonhuman primate's influence on the yellow fever transmission to humans[18].

The main objective of this research is to assess where the risk of yellow fever infections in humans could be being amplified by the contribution of enzootic cycles based on primate assemblages and a pool of sylvatic mosquito vectors. Given the current context of yellow fever geographic spread, we address this challenge through a temporal stratification of disease cases (Fig. 1), thus assuming that the current risk is a combination of past trends and the arrival of new factors.

## Results

**Zoogeographic Factor**. A total of 27 significant primate chorotypes were detected: 13 in Africa and 14 in South America (Supplementary Figs. 1 and 2). The distribution of yellow fever cases during the late 20th century was significantly related to three American chorotypes located in Brazil and Peru and two African chorotypes mostly located south of the Sahara desert and north of the Equator (Fig. 2a). In the case of the early 21st century, the distribution of yellow fever cases was significantly related with chorotypes distributed throughout most of the tropical area of South America and Africa (Fig. 2a and Supplementary Table 1).

**Baseline disease models**. All the baseline disease favourability models fitted the observed distribution of yellow fever cases, according to Hosmer & Lemeshow's test of goodness of fit (Supplementary Table 2). Trend-surface variables (i.e., the spatial autocorrelation caused by the historical patterns of the disease) characterized to the highest extent the distribution of yellow fever cases in Africa and America between 1970 and 2000, and also between 2001 and 2017 (Supplementary Table 2). The closeness to population centres was the second most important factor in explaining the distribution of cases in the late 20th century and their spread during the early 21st century. In the late 20th century, the presence of yellow fever cases was also favoured by higher slopes and annual precipitations, whereas high maximum temperatures in the warmest month became significantly relevant during the early 21st century. Among the chorotypes that were significantly related to the distribution of yellow fever cases, three chorotypes from South America were included in the late 20th-century disease model: SA7, SA8, and SA14. Four more

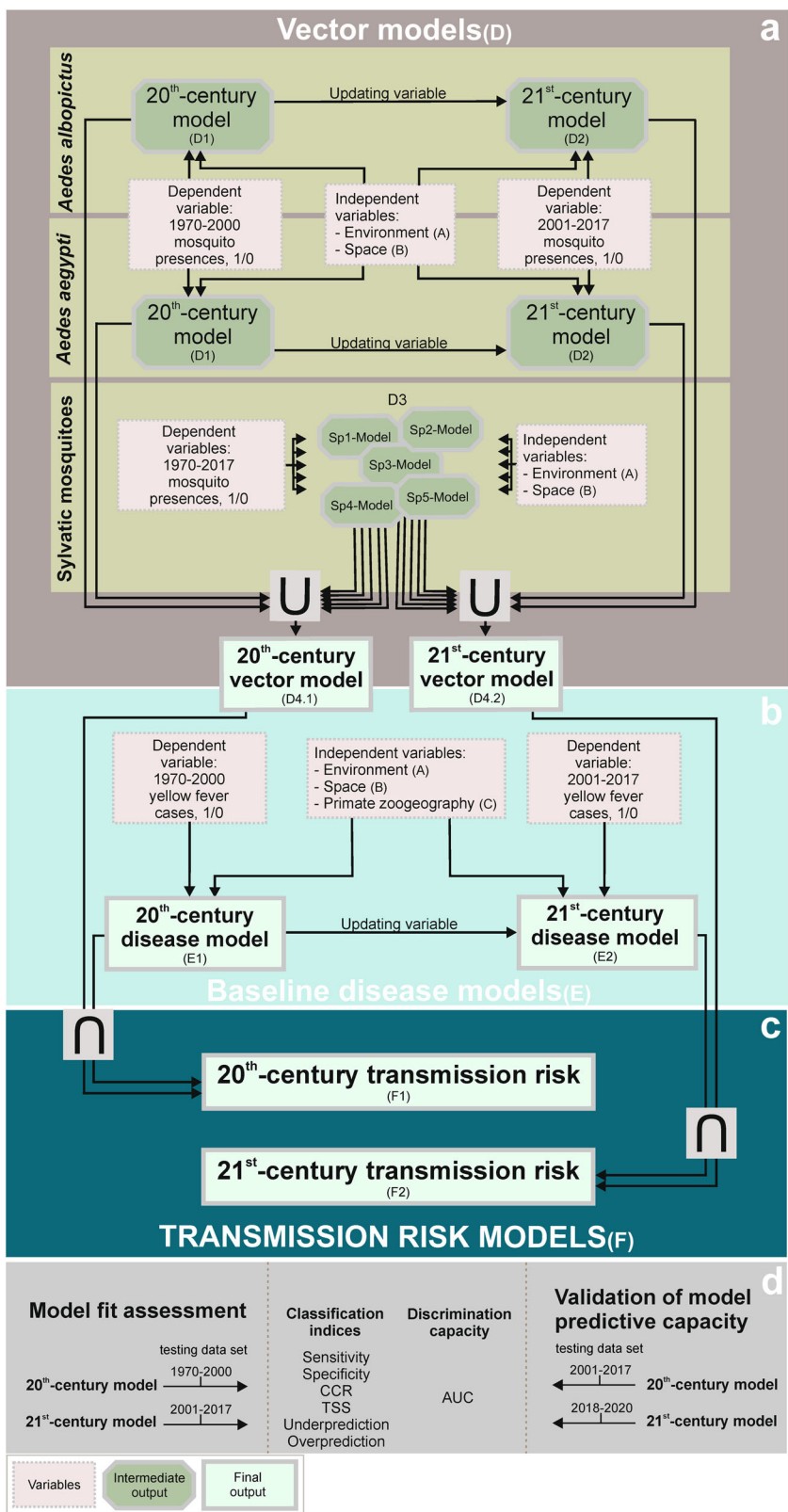

**Fig. 1 Methodological steps of the approach used for yellow fever transmission risk modelling. a** Vector models result from combining, through the fuzzy union (∪), favourable areas for the presence of urban (*Ae. aegypti* and *Ae. albopictus*) and sylvatic vectors. **b** Baseline disease models describe the areas favourable to the occurrence of yellow fever cases. **c** Transmission risk models quantify the level of yellow fever transmission risk, according to the fuzzy inteserccion (∩) between vector and baseline disease models. **d** Model fit assessment and validation of model predictive capacity. Methodological details are given in Supplementary Methods, which includes very detailed methodological explanations for all elements with a code in parentheses: A, B, C, D1, D2, D3, D4.1, D4.2, E, E.1, E.2, F, F.1, F.2.

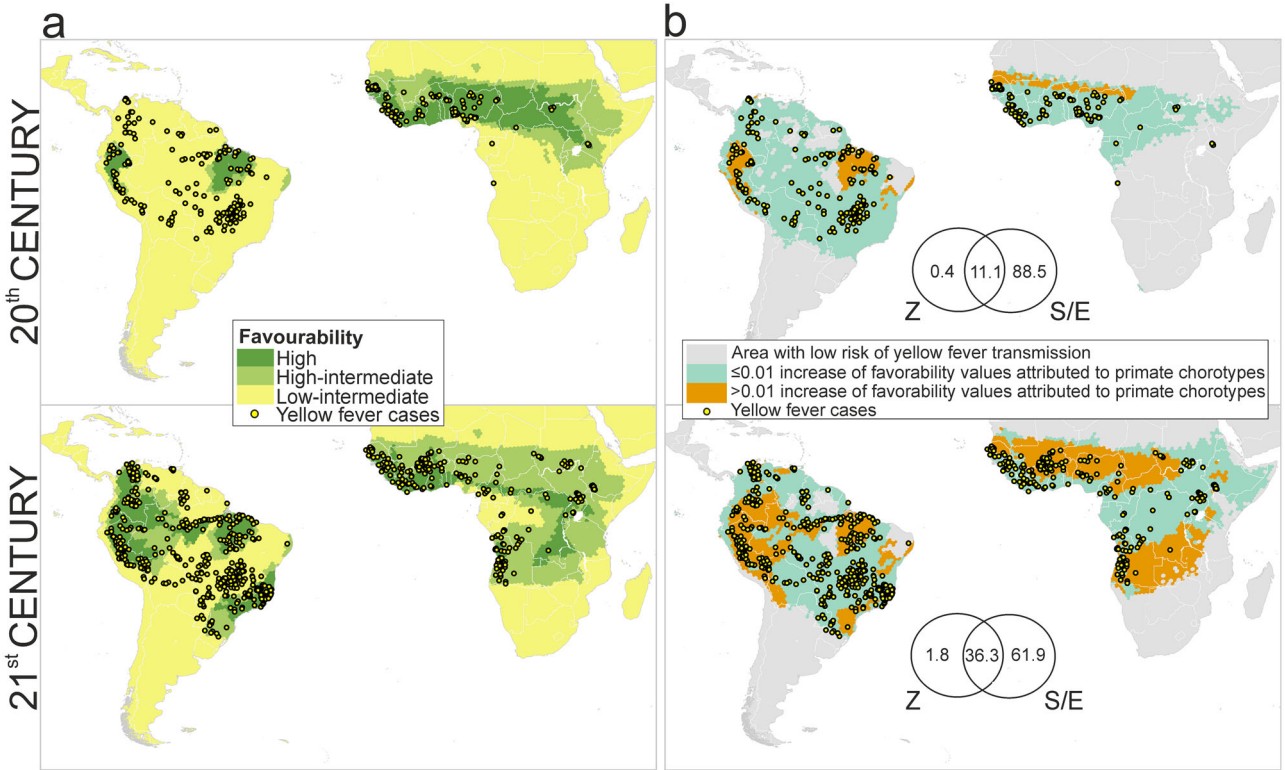

**Fig. 2 Contribution of the zoogeographic factor. a** Model of favourability for the occurrence of yellow fever cases according to the presence of non-human primate chorotypes (i.e., zoogeographic model) [the scale for favourability values is: high ($F > 0.8$); high-intermediate ($0.5 \le F \le 0.8$); low-intermediate ($0.2 \le F < 0.5$)]. **b** Partial contribution of primates on the presence of yellow fever cases in humans [the numbers are percentages of contribution to the distribution of favourability in the disease models (Z: zoogeographic factor, S/E: spatial/environmental factor)]. The maps in **a** represent the areas where the primate presence could favour the occurrence of disease cases in humans, although correlations with other factors influencing the primate biogeography (such as climate, topography or land cover) might be involved in this relation. Instead, the maps in **b** highlight the areas where the presence of primates could favour the occurrence of yellow fever regardless of correlations with other factors.

chorotypes helped to build the early 21st-century disease model: SA1, SA6, and SA12 from South America, and AF9 from Africa.

Spatial, environmental and zoogeographically favourable areas for the presence of yellow fever cases have spread southward in South America, reaching southern Paraguay, the Misiones province in Argentina, and the Atlantic forests of southern Brazil, locally called "Mata Atlantica"; and south and eastward in Africa, reaching Namibia, Zambia, Tanzania, Kenya and Somalia (Fig. 3).

**Relative importance of the zoogeographical factor**. The pure contribution of primate chorotypes to explaining the distribution of yellow fever cases in the late 20th century is 0.4%. Nevertheless, chorotypes could explain up to 11.5%, because 11.1% of the variation in favourability can be as much attributed to the presence of primate chorotypes as to the spatial/environmental factor (Fig. 2b). These percentages are higher in the early 21st-century model: 1.8% for the pure contribution of primate chorotypes, and 38.1% for the intersection between chorotypes and the spatial/environmental factor (Fig. 2b).

In the late 20th century, primates might have had a relevant role in explaining the occurrence of yellow fever cases in humans in easthern Brazil, northern Peru, and western Sahel (Fig. 2b). The area of influence of primates seems to have spread during the early 21st century, expanding northward, eastward and southward in Peru, Colombia, Venezuela, Ecuador and very prominently in Brazil, where it reached the "Mata Atlantica". In Africa, the spread of the primate contribution might have affected most of the tropical regions, only excluding the rainforest domain

(Fig. 2b). These areas include countries of West Africa, Angola, south of the DRC, and some zones in Tanzania and Kenia.

**Vector models**. Although the distribution of yellow fever cases is restricted to Africa and South America, there are favourable areas for the presence of vector species in all continents except Antarctica, as outlined by the fuzzy union of all the single-mosquito-species models performed (Fig. 3). During the last two decades, spatial and environmentally favourable conditions seem to have spread northward in Europe, USA, and China; eastward in Central Africa; and south and westward in South America. In contrast, favourability has decreased in Oceania and Japan.

**Yellow fever transmission-risk model**. In the early 21st century, the risk of yellow fever transmission in South America could have increased in Paraguay, in some provinces of northern Argentina (e.g., Misiones and Corrientes), and in the "Mata Atlantica" in the south-east of Brazil (Fig. 3). The spread of *Aedes* mosquitoes in central Brazil might have also increased the level of risk in the region between Mato Grosso and Bahia states. Similarly, the increase of the transmission-risk in Africa resembles the south and westward spread of favourable areas for the presence of yellow fever cases (Fig. 3).

The enhancement of the early 21th-century transmission-risk model, through the use of variables only available for this period, provided little but meaningful changes compared to the original transmission-risk model (Figs. 3 and 4). In America, the enhanced model pointed to a moderate level of risk in the north-eastern states of Brazil. In Africa, the highest levels of

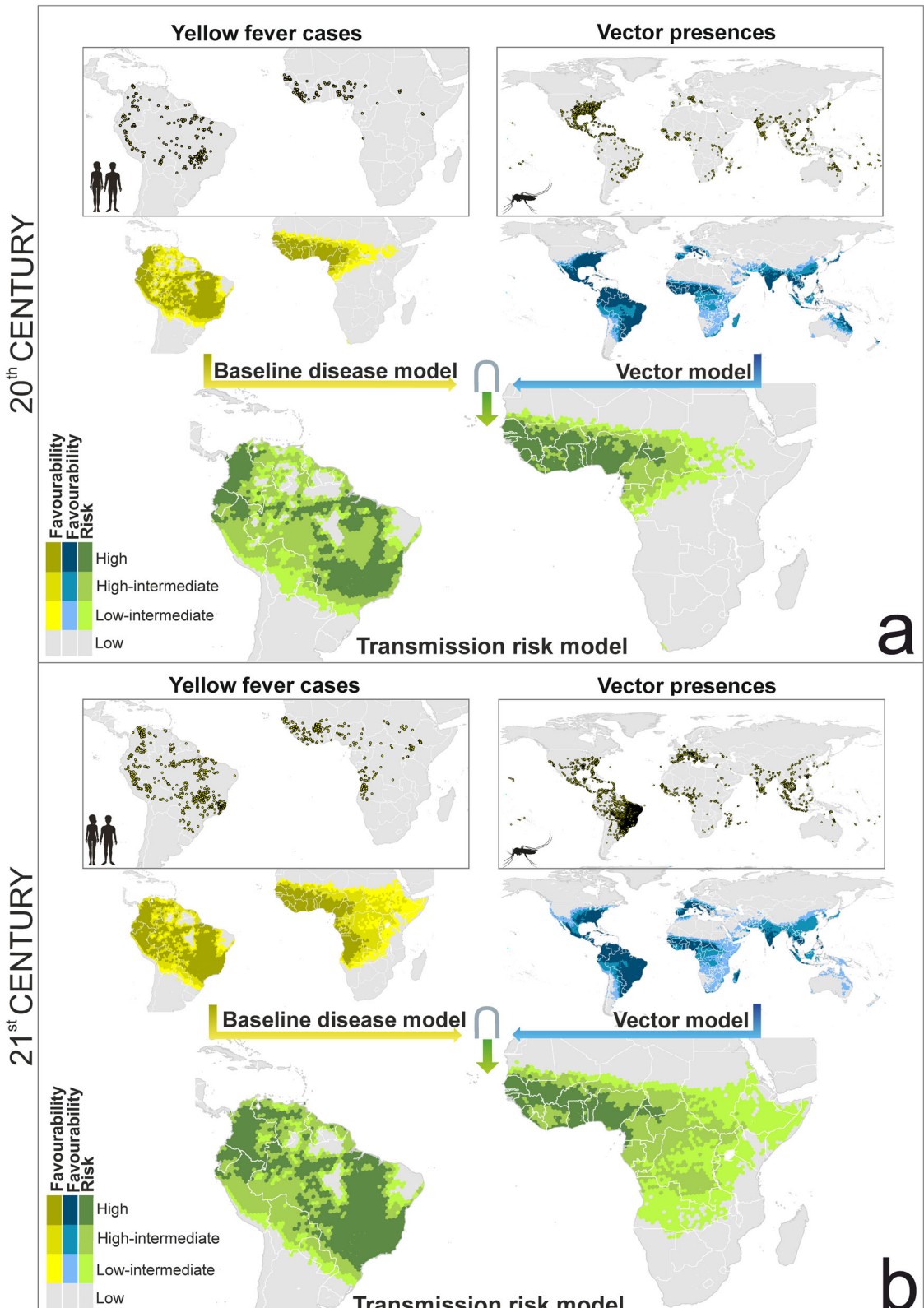

**Fig. 3 Global baseline disease, vector and transmission-risk model. a** maps for the late 20th century. **b** maps for the early 21st century. The risk of transmission is estimated as the fuzzy intersection (∩) between favourable conditions for the occurrence of yellow fever cases, and favourable conditions for the presence of vector species. The Favourability values were considered on the following scale: High (F > 0.8); High-Intermediate (0.5 ≤ F ≤ 0.8); Low-Intermediate (0.2 ≤ F < 0.5); and Low (F < 0.2). The spatial resolution is based on 7,774-km$^2$ hexagons. Recorded occurrences of yellow fever cases and of vector presences are also mapped (see "Yellow fever datasets" and "Vector dataset" in the Methods section for details). Mosquito and humans clip art source: http://www.freepik.com.

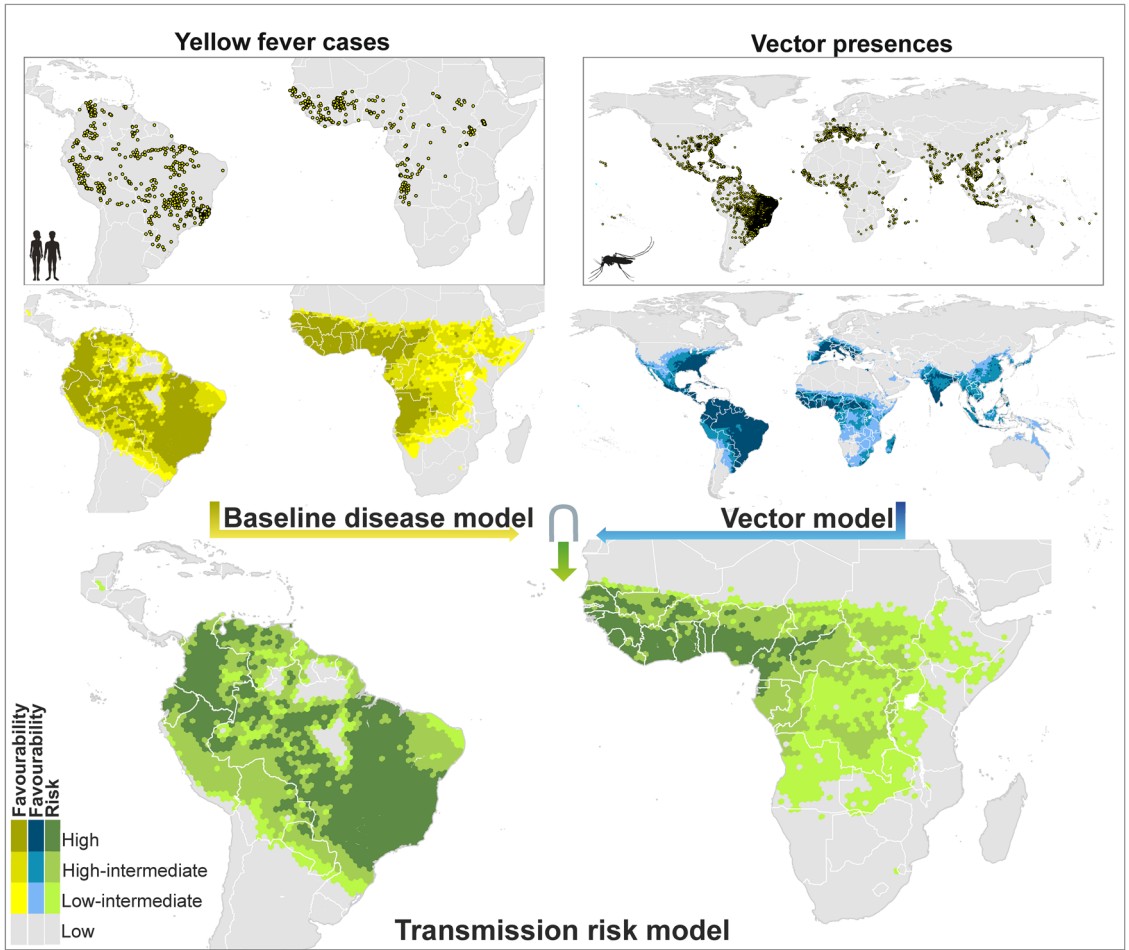

**Fig. 4 Enhanced global baseline disease, vector and transmission-risk models for the early 21st century.** The risk of transmission is estimated as the fuzzy intersection (∩) between favourable conditions for the occurrence of yellow fever cases, and favourable conditions for the presence of vector species. Compared to the models in Fig. 3, additional predictor variables only available for the 21st century were considered. Recorded occurrences of yellow fever cases and of vector presences are also mapped. The Favourability values were considered on the following scale: High (F > 0.8); high-intermediate (0.5 ≤ F ≤ 0.8); Low-Intermediate (0.2 ≤ F < 0.5); and Low (F < 0.2). The spatial resolution is based on 7774- km² hexagons. Recorded occurrences of yellow fever cases and of vector presences are also mapped (see "Yellow fever datasets" and "Vector dataset" in the Methods section for details). Mosquito and humans clip art source: http://www.freepik.com.

transmission risk were mostly concentrated in countries of the Atlantic coast extending from the Sahel to the Equator. Here, in contrast to the not-enhanced model, intermediate-risk levels hardly reach the Horn of Africa, which coincides with the record of yellow fever cases and vector presences.

**Model fit assessment.** Discrimination and classification index values are shown in Table 1. All the models showed an outstanding discrimination capacity according to Hosmer and Lemeshow (2000) (i.e., the area under the receiver operator characteristic curve, AUC, was always >0.91). In all models, true skill statistic (TSS) values were positive and >0.582. In the disease and transmission-risk models, correct classification rate (CCR) values were between 0.83 and 0.91 depending on the favourability threshold considered, and it was between 0.61 and 0.78 in the vector models. Sensitivity values indicate that >85% (and except in two cases, >90%) of hexagons with vector presences and yellow fever cases reported were correctly classified by the models, regardless of the favourability threshold; and underprediction values were never >0.004. So, the occurrence of vectors and yellow fever cases reported, and finally the risk of yellow fever transmission, were rarely underestimated. Specificity values were between 0.82 and 0.91 in the disease and transmission-risk

models, and between 0.57 and 0.76 in the vector models. Finally, overprediction values were in the range 0.70-0.92, i.e., a high proportion of hexagons considered favourable for the presence of disease cases or vectors have not been reported to have them. This means that the scattered appearance of presence data did not generate geographically scattered favourability models (see Figs. 3 and 4), and so that these models were far from overfitting.

**Validation of model predictive capacity.** Model's discrimination and classification capacities remained quite similar or increased, compared to those in Table 1, when disease cases regarded to "future information" (i.e., data corresponding to time periods later than those considered for model training) were included in the assessment (see Table 2). In the late 20th-century transmission-risk models, the use of yellow fever cases from 2001-2020 led to sensitivity values decreased from 0.922–0.982 (Table 1) to 0.791–0.864 (Table 2) and specificity values increased from 0.876–0.909 (Table 1) to 0.885–0.917 (Table 2).

Using yellow fever cases from 2018 to 2020 for the assessment of the early 21st-century transmission-risk models, sensitivity values increased from 0.876–0.992 (Table 1) to 0.904–1 (Table 2) and specificity values decreased from 0.833–0.894 (Table 1) to 0.815–0.877 (Table 2). The fact that sensitivity of the

**Table 1 Model fit assessment based on discrimination and classification capacities respect to vector and disease records of the same period.**

|  | Model | AUC | FCT | Sens. | Spec. | CCR | TSS | Underp. | Overp. |
|---|---|---|---|---|---|---|---|---|---|
| Late 20th century | Disease | 0.971 | 0.5 | 0.959 | 0.905 | 0.905 | 0.864 | 0.001 | 0.895 |
|  |  |  | 0.2 | 0.982 | 0.873 | 0.875 | 0.855 | 0.000 | 0.917 |
|  | Vectors | 0.919 | 0.5 | 0.946 | 0.705 | 0.717 | 0.651 | 0.004 | 0.849 |
|  |  |  | 0.2 | 0.991 | 0.591 | 0.612 | 0.582 | 0.001 | 0.881 |
|  | Transmission risk | 0.962 | 0.5 | 0.922 | 0.909 | 0.909 | 0.831 | 0.001 | 0.895 |
|  |  |  | 0.2 | 0.982 | 0.876 | 0.877 | 0.858 | 0.000 | 0.915 |
| Early 21st century | Disease | 0.963 | 0.5 | 0.961 | 0.872 | 0.874 | 0.833 | 0.001 | 0.832 |
|  |  |  | 0.2 | 0.988 | 0.826 | 0.830 | 0.814 | 0.000 | 0.868 |
|  | Vectors | 0.928 | 0.5 | 0.947 | 0.740 | 0.761 | 0.687 | 0.008 | 0.715 |
|  |  |  | 0.2 | 0.994 | 0.576 | 0.618 | 0.570 | 0.001 | 0.796 |
|  | Transmission risk | 0.951 | 0.5 | 0.876 | 0.889 | 0.889 | 0.765 | 0.004 | 0.825 |
|  |  |  | 0.2 | 0.980 | 0.833 | 0.837 | 0.813 | 0.001 | 0.864 |
| Early 21st century (enhanced model) | Disease | 0.964 | 0.5 | 0.968 | 0.872 | 0.875 | 0.840 | 0.001 | 0.831 |
|  |  |  | 0.2 | 0.994 | 0.833 | 0.837 | 0.827 | 0.000 | 0.863 |
|  | Vectors | 0.932 | 0.5 | 0.958 | 0.752 | 0.772 | 0.710 | 0.006 | 0.704 |
|  |  |  | 0.2 | 0.991 | 0.616 | 0.653 | 0.607 | 0.002 | 0.780 |
|  | Transmission risk | 0.954 | 0.5 | 0.878 | 0.894 | 0.894 | 0.772 | 0.004 | 0.818 |
|  |  |  | 0.2 | 0.992 | 0.843 | 0.847 | 0.835 | 0.000 | 0.855 |

*AUC* area under the receiver operator characteristic curve, *FCT* favourability classification threshold, *Sens.* sensitivity, *Spec.* specificity, *CCR* correct classification rate, *TSS* true skill statistic, *Underp.* underprediction rate, *Overp.* overprediction rate.

**Table 2 Validation of model predictive capacity based on discrimination and classification performance respect to yellow fever records of a later period.**

|  | Records of reference | Model | AUC | FCT | Sens. | Spec. | CCR | TSS | Underp. | Overp. |
|---|---|---|---|---|---|---|---|---|---|---|
| Late 20th century | 2001 to 2020 | Disease | 0.936 | 0.5 | 0.805 | 0.913 | 0.911 | 0.718 | 0.006 | 0.800 |
|  |  |  |  | 0.2 | 0.864 | 0.883 | 0.883 | 0.747 | 0.004 | 0.834 |
|  |  | Transmission risk | 0.933 | 0.5 | 0.791 | 0.917 | 0.914 | 0.708 | 0.006 | 0.795 |
|  |  |  |  | 0.2 | 0.864 | 0.885 | 0.885 | 0.749 | 0.004 | 0.832 |
| Early 21st century | 2018 to 2020 | Disease | 0.939 | 0.5 | 0.932 | 0.853 | 0.854 | 0.785 | 0.000 | 0.976 |
|  |  |  |  | 0.2 | 1.000 | 0.808 | 0.809 | 0.808 | 0.000 | 0.980 |
|  |  | Transmission risk | 0.938 | 0.5 | 0.932 | 0.873 | 0.873 | 0.805 | 0.000 | 0.972 |
|  |  |  |  | 0.2 | 1.000 | 0.815 | 0.816 | 0.815 | 0.000 | 0.979 |
| Early 21st century (enhanced model) | 2018 to 2020 | Disease | 0.937 | 0.5 | 0.904 | 0.853 | 0.853 | 0.757 | 0.000 | 0.977 |
|  |  |  |  | 0.2 | 0.986 | 0.814 | 0.815 | 0.800 | 0.000 | 0.980 |
|  |  | Transmission risk | 0.936 | 0.5 | 0.904 | 0.877 | 0.877 | 0.781 | 0.000 | 0.972 |
|  |  |  |  | 0.2 | 0.986 | 0.825 | 0.825 | 0.811 | 0.000 | 0.979 |

*AUC* area under the receiver operator characteristic curve, *FCT* favourability classification threshold, *Sens.* sensitivity, *Spec.* specificity, *CCR* correct classification rate, *TSS* true skill statistic, *Underp.* underprediction rate, *Overp.* overprediction rate.

transmission-risk model increases, only with these "future" values, means that 90–100% of cases recorded after the model-training period were reported in areas predicted to be at risk of yellow fever transmission to humans.

## Discussion

The most relevant contribution of our results is the mapping of the areas where zoonotic cycles (involving primates and sylvatic mosquitoes) could be currently favouring the occurrence of yellow fever virus transmission to humans, based on the most updated data-base of humans cases. In South America, these areas include wide regions within the western, eastern and central Amazon basin, and also great part of the "Mata Atlantica", that is the Atlantic forests of Brazil. In Africa, they largely overlap with the open and forested savannas to the north and south of the Central African rainforests. This might be, however, a conservative view of the geographic relevance of the yellow fever zoonotic cycle, as only the areas where the primate contribution to risk was not correlated with environmental factors were

mapped. During the late 20th century, the influence of zoonotic cycles on yellow fever outbreaks could have been restricted to the Peruvian and the easternmost Brazilian Amazon in America, and to the southern limits of the Sahel in Africa. So, the understanding of this dynamic relationship between primates and yellow fever needs to be put in the context of recent changes in the global distribution of reported infection in humans.

Many yellow fever cases have been recorded in the early 21st century in areas that were free from the disease during the late 20th century. For example, disease cases are currently reported in south and southeastern Brazil (Fig. 3), where no cases seem to have occurred for decades before 2005[13]. According to our baseline disease models, favourability for yellow fever transmission in southeastern Brazil was intermediate ($0.2 \leq F \leq 0.8$) in the late 20th century, and high ($F > 0.8$) in the early 21st century (Fig. 3). This increase in favourability could be a consequence of a modern-virus-lineage arrival from the north of South America[13], that has caused all major yellow fever outbreaks in the subcontinent since 2000, including those in Brazil in 2008 and 2016 (details are discussed below).

In Africa, the areas favourable to yellow fever transmission are currently spreading from the west of the continent to the south and east, reaching countries such as the DRC, Angola, Namibia, Zambia, Tanzania, and Somalia (Fig. 3). In 2015 and 2016, the most widespread outbreaks reported in Africa in more than 20 years took place in Angola and the DRC[29]. Several factors might be responsible for the lower yellow fever activity in East and Central Africa before 2000. According to Mutebi et al.[30], genetic differences between yellow fever genotypes may play an important role in the distribution pattern of yellow fever outbreaks in Africa, the West African genotypes being associated to more frequent outbreaks. In addition, the ecological diversity and behaviour of vector mosquito species may also influence on the African yellow fever biogeography. In West Africa, *Ae. Aegypti* populations are responsible for urban outbreaks, whereas no yellow fever cases have been attributed to this species in East Africa[30]. This applies, for example, to the 1992–1993 outbreak happened in Kenya[30]. What is interesting is that *Ae. aegypti* occurs in both East and West Africa, but there are two subspecies: *Ae. ae. aegypti* and *Ae. ae. formosus*. Crawford et al[31]. mentioned that *Ae. ae. aegypti* is a demonstrated vector of yellow fever, whereas the extent to which *Ae. ae. formosus* lives alongside and feeds on humans is unclear[31]. These authors suggest that the distribution of *Ae. ae. aegypti* is likely limited to West Africa, eastern populations of the species probably belonging to *Ae. ae. formosus*. So, it is worth investigating whether the distribution of viral and vector genetic variants is currently changing together with the distribution of yellow fever reports, or whether the apparent geographic spread of virus infections is a result of increased reporting efforts.

The interpretation of models based on incomplete information must be made with caution. Sampling bias could lead to interpretation mistakes if the model output maximized the geographic fit between recorded disease cases and the resulting predictions. The combination of logistic regressions and the favourability function, used to get the outputs of our models, do not usually tend to overfitting[32], hence these models showed high overprediction rates (Table 1), that is, a high proportion of favourable areas that are not recorded to have had cases. Part of this overprediction might be explained by transmission risks forecasted in areas prone to be endemic in the short term, as shown by the fact that the overprediction rate of the 20th-century model was around 0.9 (Table 1), but it decreased until about 0.8 when this model's prediction capacity was tested according to the 21st-century cases (Table 2). However, our model outputs are also consistent with the belief that the reporting of yellow fever cases could underestimate the actual number of cases, this being up to 500 times higher than reported in Africa, and around 10 times higher in South America[6,33]. Anyway, there are limits for any kind of model trying to analyse the geography of infection patterns for a virus showing high spatio-temporal dynamism, that is affected by the emergence of new lineages, by the changing distribution of *Aedes* mosquitoes[34], as much as by revisions of vaccination strategies. So, our models have to be interpreted in the current historical context, as they were designed for this specific spatio-temporal window.

As expected, our 21st-century model outputs resemble the results of recent studies focused on the distribution of yellow fever risk areas. Gaythorpe et al.[28] evaluated the vaccine effectiveness in South America and Africa with a province-level model of the probability of yellow fever case reporting. Compared to this model, our outputs show a similar geographic pattern of yellow fever transmission risk; however, we point to higher risk in South and West Brazil and other South American countries such as Venezuela. Our 21st-century model also outlines the presence of high transmission risk in African countries not highlighted by

Gaythorpe et al.[28], such as Zambia and Tanzania. In Tanzania, while no yellow fever cases have been reported for decades[35], a study published by Rugarabamu et al.[36] provides evidence of yellow fever exposure, suggesting the need to strengthen the surveillance system. A later research by Shearer et al.[6] assessed the pattern of the speed at which human individuals are expected to acquire yellow fever virus infection in a given location. In principle, Shearer et al.'s results are not comparable to our risk maps because these authors used the estimated area at risk of yellow fever infections[27] as geographic extent for their analyses. Nevertheless, their map of "individual apparent infectious risk" points to areas partially related to those in which non-human primates seem to increase the risk of yellow fever transmission to humans (Fig. 2b). This happens, for example, in regions of the Amazon Basin in Peru, Ecuador, Colombia, Venezuela, and North and East Brazil; and of Africa in the southern limits of the Sahel. The overlap could be even higher because, as mentioned above, we present a conservative view of the zoonotic-cycle influence area. Taking into account that Shearer et al. considered the distribution of primates as predictor variables in their model (together with other environmental factors), we wonder whether, in large regions of South America and Africa, the risk of human individual infection[6] could be linked to the increased risk of transmission favoured by non-human primates (Fig. 2). The most relevant difference between Shearer et al.'s maps and ours is located in Southeastern Brazil, probably because they did not considered reports of the most recent yellow fever outbreaks in the area (which happened after 2015).

Deep surveillances should be encouraged in primates of southern Brazil given the active evolutionary dynamism experienced by the yellow fever virus in South America. In this continent, a yellow fever virus "modern lineage" has spread since 1989 out of the endemic areas[13]. From 1980 to the middle of 2015, 792 sylvatic yellow-fever cases in humans and 421 deaths were reported in Brazil[37]. The modern lineage has diversified into several sub-lineages, one of which seems to be responsible of the disease re-emergence in 2016–2019 in the "Mata Atlantica" of the southern states of Brazil, causing one of the largest epizootic outbreaks recorded in the country[13]. The enzootic maintenance in primates is believed to be the background of the yellow-fever virus evolution in America[13]. However, the modern yellow fever lineage seems to have evolved in Trinidad and Tobago[38], and the "modified" modern lineage, introduced in 2016 in Southern Brazil, was probably a result of human translocations from Venezuela[13]. Nevertheless, the involvement of primates on transmission to humans in the Mata Atlantica, and the high evolutiounary rate recently shown by the yellow fever virus, might derive in new variants that could reach human populations, taking into account that the zoonotic cycles in Southeastern Brazil are closely connected with urban areas[3].

Forest loss and fragmentation could be increasing the proximity of humans to other primates, enhancing the zoonotic transmission of yellow fever. In fact, our 21st century disease model included forest loss after 2001 as a significant predictor variable (Supplementary Table 2). Habitat destruction affects significantly the ecology of emerging infectious diseases in wildlife and humans[39,40]. In undisturbed ecosystems, pathogens are diluted in the animal community, whereas some reservoir species and their pathogens may begin to dominate in fragmented forests[40] (for example, Olivero et al. 2017[41] demonstrated that Ebola outbreaks located along the limits of the African rainforest biome were significantly associated with forest losses that took place within the previous 2 years). A study published by Cunha et al.[3] has proposed a new scenario for the 2016–2019 yellow fever outbreak in São Paulo, indicating that yellow fever sylvatic transmission among primates probably occurs in the city. The

virus has been recorded in urban titis (*Callithrix* sp.) and in mosquito species normally inhabiting the forest (e.g., in *Aedes scapularis*). Consequently, the yellow fever pathogeography could be providing a new example of how forest fragmentation could amplify the risk of disease transmission by increasing the proximity of human populations to wildlife, in this case through the occupancy of urban areas by primates and sylvatic mosquitoes.

In case yellow fever surveillances in non-human primates were addressed, we propose to focus on the list of species belonging to chorotypes significantly related to the disease distribution (Supplementary Figs. 1 and 2). These chorotypes have represented the diversity of primates reportedly infected by the yellow fever virus so in South America as in Africa. In Brazil, during the period 1996 to 2016, 2221 deaths in non-human primates caused by yellow fever virus were reported, whereas only in the 2016–2019 outbreak the number of recorded deaths was 3569[37]. In this lastest outbreak, yellow fever virus infections were detected in specimens of genera *Alouatta*, *Brachyteles*, *Callicebus*, *Callithrix*, *Leontopithecus* and *Sapajus*, all of them inhabiting the Mata Atlantica[37,42,43]. All these genera include species whose biogeographic patterns characterize significantly the distribution of humans yellow fever cases in southern Brazil (i.e., species belonging to chorotypes SA4 and SA11, see Supplementary Fig. 1). Overall in South America, according to the available literature[6,37,42,43], the yellow fever virus has been detected in 13 nonhuman primate genera, all of which include species whose biogeography is related to the American distribution of humans cases (see Supplementary Fig. 1). The same applies in Africa, where the yellow fever virus has been detected in 11 non-human-primate genera (mostly in *Cercocebus*, *Cercopithecus*, *Colobus*, *Erythrocebus*, *Galago*, *Otolemur*, *Papio*, *Perodicticus* and *Pilocolobus*)[6,37,42,43]. All these genera include species belonging to the chorotypes that are significantly related to the yellow fever distribution (see Supplementary Fig. 2).

Our analyses can contribute to identify new areas that should be prioritised for vaccination, for which we propose to take three yellow fever transmission geographic scenarios into account: (1) areas with very favourable conditions (F ≥ 0.5) for both the presence of the virus and mosquito vectors, in which case the risk of transmission is very high; (2) areas with low but not negligible risk of yellow fever transmission (0.2 ≤ F ≤ 0.5); and (3) areas environmentally favourable to the presence of mosquito vectors, but not to the virus occurrence. In South America, the first and most severe scenario occurs in southern Brazil. Following the 2016−2019 outbreak, the WHO has programmed vaccination in this area[44]. Our models provide support for the additional vaccination programme planned by the Brazilian Health Ministry in 2019[44] in eastern Brazil (involving states such as Pernambuco, Alagoas, Paraíba, Sergipe, and Ceará, see Fig. 4), despite vaccination in this area is not yet considered by the WHO[44] and the CDC[45] to be a priority. The African areas included in the most severe scenario are located in west and central countries of the continent, where the WHO already suggests prescriptive vaccination[46,47]. A set of African areas fits the scenario with low but not negligible risk of yellow fever transmission, but are not yet considered for vaccination by the WHO[47] and the CDC[45]: the north of Namibia, the west of Zambia, the east of Ethiopia, and some areas in Somalia (Fig. 4). We propose that active yellow fever surveillance strategies be considered for these areas in order to be alert for outbreaks in the near future. Finally, mosquito vectors already occur or find favourable conditions in many areas in North America, southern Europe, Asia and Oceania that are outside the yellow fever endemic area. The most suitable policies in these cases may involve preventing virus introduction by international travelling. For this reason, there are countries that require the yellow fever vaccination certificate for travellers[48],

although this certificate is not required in some countries with high-risk zones according to our models that also coincide with areas in which vaccination is recommended by the WHO[48]. A global strategy could be designed for granting, with no exceptions, vaccination of third-country citicens entering in countries at high or medium risk of yellow fever transmission[48]. In addition, vaccination should be also considered an option in areas with stable vector-mosquito populations that are close to the endemic areas, as is the case of Uruguay, northern Argentina and the eastern coast of Africa. Vaccination campaigns recommended by the WHO in the provinces of Misiones and Corrientes[44], to the north of Argentina, are positive examples of this kind of initiatives.

## Methods

**Lattice data geoprocessing and temporal extent.** We latticed the data[49] using a worldwide grid composed of 18,874 hexagonal 7774 km² units, built using Discrete Global for R (https://github.com/r-barnes/dggridR)[50]. All the information we processed on yellow fever cases, on urban and sylvatic vectors presences, and on zoogeographic, spatial and environmental variables (see details on this information below) was aggregated at this spatial resolution. We used zonal statistics to calculate average variable values using ArcMAP 10.7.

The temporal extent for our analysis was divided into three periods: the late 20th century (1970–2000), the early 21st century (2001–2017), and the period 2018–2020. Predictions estimated by the late 20th century models were validated using cases reported in the early 21st century, and predictions from the early 21st century models were validated with records from 2018-2020. Although the limit between periods at the turn of the century is arbitrary, it reflects: 1) Distributional changes in the ranges of the *Ae. aegypti* and *Ae. albopictus* vectors[51]; 2) after 1999, the yellow fever genotype I has spread outside the endemic regions, and the genotype I modern-lineage has caused all major yellow fever outbreaks detected in non-endemic regions of South America since 2000[13]; 3) the maximum potential of globalization was realised at the beginning of the 21st century with the opening of international borders, the widespread access to the Internet and to cell phones, and the generalization of online travel booking and of low-cost flights[34]. The end of the second period, 2017, was chosen in order to include three years with occurrence of yellow fever cases in south-western Brazil (and two since its occurrence in Angola and the DRC), while retaining three later years for predictive testing purposes (details on this testing are given below).

**Yellow fever datasets.** We used georeferenced cases of yellow fever in humans for a period of 51 years (from 1970 to 2020). This study period starts immediately after the suspension of the use of DDT due to to the appearance of resistance of *Ae. aegypti* in the late 1960s in several countries, after 50 years of eradication efforts[10]. We took from Shearer et al.[6] the distribution of yellow fever cases for the period 1970–2016. We extracted additional cases for the period 1970–2020 from various sources (Supplementary data 1), including ProMED-mail: Program of International society for infectious diseases; World Health Organization (WHO): Yellow fever outbreak weekly situation reports, Rapport de situation fievre jaune en RD Congo and Weekly epidemiological record; Health Ministry of different countries: Epidemiological Bulletins of yellow fever in Brazil, Peru, Colombia, and Paraguay; Pan American Health Organization (PAHO): Epidemiological Update Yellow Fever; European Centre for Disease Prevention and Control (ECDC): Communicable disease threats report and Rapid risk assessment report; Nigeria Centre for Disease Control (NCDC): Situation report, yellow fever outbreak in Nigeria and Global Infectious Disease and Epidemiology Online Network (GIDEON). The reported cases were complemented with publications available since 2016 with georeferenced information on case location (Supplementary data 1). In addition, information was also sought on cases reported in French and Portuguese from local news reports in Africa.

We only represented in the hexagonal lattice the reported cases of yellow fever that had a precise location or that were referred to administrative unit was smaller than or of similar size to the hexagons. This dataset was transformed into a binary variable per study period representing the presence (n = 218 hexagons in the late 20th century; 493 hexagons in the early 21st century, see Supplementary data 2) or absence (n = 18,656 hexagons in the late 20th century; 18,381 hexagons in the early 21st century), hereafter the distribution of reported cases of yellow fever.

**Vector dataset.** The georeferenced presences of vectors involved in the urban cycle of yellow fever (i.e., the mosquitoes *Ae. aegypti* and *Ae. albopictus*) were taken from "The global compendium of the *Ae. aegypti* and *Ae. Albopictus* occurrence"[26] for the period 1970–2014. We complemented these records with georeferenced data scientifically validated for the period 2014–2017, taken from VectorBase (https://www.vectorbase.org/) and Mosquito Alert (http://www.mosquitoalert.com/). We included both species because, although *Ae. Aegypti* is the main vector of yellow fever, *Ae. albopictus* can also transmit the yellow fever virus to humans[4,52].

In addition, we included georeferenced occurrence data of sylvatic vectors (*Haemagogus janthinomys*, *H. leucocelaenus* and *Sabethes chloropterus* in South

America; *Ae. africanus* and *Ae. vittatus* in Africa), which were obtained from Vectormap (vectormap.si.edu) and Gbif (https://gbif.org).

We represented in the hexagonal lattice the reported occurrence of mosquitoes that had a precise location or were located in administrative smaller than or of similar size to the hexagons. With this information, we built binary variables representing the presence or absence of each mosquito species in each hexagon. For species involved in the urban cycle, we built two binary variables per species: one for the late 20th century, and another for the early 21st century. For species involved in the sylvatic cycle, we merged the data of late 20th century and early 21st century in order to build a binary variable per species, due the scarcity of data and under the assumption that their distributions have been stable during the four last decades[53–55].

**Zoogeographic, spatial and environmental variables**. We built zoogeographic variables based on chorotypes, or types of distribution ranges, of all non-human primate species, as all are potentially vulnerable to yellow fever[56]. A chorotype is a distribution pattern shared by a group of species[57]. For obtaining these zoogeographic variables, we proceeded in 4 steps: (1) Distribution maps of non-human primates were obtained from the IUCN for South-America and Africa; (2) the species ranges were classified hierarchically using the classification algorithm UPGMA according to the Baroni-Urbani & Buser´s similarity index[58]; (3) we evaluated the statistical significance of all clusters obtained as a result of the classification using RMacoqui 1.0 software (http://rmacoqui.r-forge.r-project.org/)[59]; (4) in each hexagon, the number of species belonging to each chorotype was quantified. We generated a zoogeographic model based on the non-human primates chorotypes by running a forward-backward stepwise logistic regression using presence/absence of yellow fever cases and the number of species of each chorotype as dependent and predictor variables, respectively. This procedure was made for two periods: late 20th century and early 21st century. Henceforth, only the selected chorotype variables were considered in the baseline disease favourability models explained below.

We built a yellow fever spatial variable for each continent (South-America and Africa), which were calculated through the trend surface approach, by performing a backward-stepwise logistic regression of the distribution of yellow fever cases on an ensemble of variables defined for polynomial combinations of longitude ($X$) and latitude ($Y$) up to the third degree: $X$, $Y$, $XY$, $X^2$, $Y^2$, $X^2Y$, $XY^2$, $X^3$, and $Y^3$. We transformed probability values derived from logistic regression into spatial favourability values by applying the Favourability Function[60,61], using the following equation:

$$F = \frac{P}{1-P} \bigg/ \left( \frac{n_1}{n_0} + \frac{P}{1-P} \right) \qquad (1)$$

where $P$ is the spatial probability of occurrence of at least a case of yellow fever at each hexagon, and $n_1$ and $n_0$ are the numbers of hexagons with presence and absence of yellow fever cases, respectively. We built a different spatial variable for each continent and time period.

We used environmental variables related to the following factors: climate, human activity, topography, hydrography, biome, ecosystem type, and forest loss. For details about the source and description of the environmental variables selected, see Supplementary Table 3.

**Pathogeographical approach to transmission risk modelling**. Our objectives were to construct a global yellow fever transmission risk map, and to identify areas where primates contribute to increased risk, using the methodology previously used to analyse the worldwide dynamic biogeography of zoonotic and anthroponotic dengue[34] (see flowchart in Fig. 1 and Supplementary Methods). We produced a transmission model focused on the late 20th century and another for the early 21st century.

The risk of transmission was assessed by combining a first model describing areas favourable to the presence of yellow fever, i.e., the "baseline disease model"; and another model describing areas favourable to the presence of mosquitoes known to act as vectors, i.e., the "vector model". For this combination, we used the fuzzy intersection[62], i.e., the risk of transmission at each hexagon was valued at the minimum between favourability in the baseline disease model and favourability in the vector model.

In this way, we considered that the vectors are a limiting factor, and that the risk of transmission derives from the degree to which the environmental conditions are simultaneously favourable for the presence of vectors and for disease cases to occur[63]. In order to analyze the spatio-temporal dynamic of yellow fever, we made comparable models for the late 20th century and the early 21st century, using predictor variables that are available for both periods. Later, we made a 21st-century enhanced model that optimized the predictive capacity of availabe information in the search for current risk areas. For this purpose, we included, in the variable set, predictors that are only accessible for the 21st century (e.g., high-resolution population density, livestock, irrigation, infrastructures, intact forest, and GlobCover land cover classes; see Supplementary Table 3).

**Baseline disease models**. The baseline disease model in the late 20th century was expressed in terms of favourability values, using the Eq. (1) (see above). This time, $P$ was calculated through a multivariable forward-backward stepwise logistic

regression of the 20th-century yellow fever presences/absences on a set of zoo-geographic, environmental and spatial variables. This was made in two blocks: 1) a stepwise selection of environmental and spatial variables; 2) a later stepwise addition of chorotypes whose presence contribute to improve significantly the likelihood of the model based only on the first block. Variables for each block were preselected using RAO´s score tests (which estimated the significance of its association to the distribution of yellow fever cases), and Benjamini and Hochberg´s (1995) false discovery rate (FDR) to control for Type I errors, which could pass due to the number of variables analysed. We also avoided excesive multicollinearity by preventing that variables with Spearman correlation values >0.8 were included in the same model. In case this happened, only the variable with the most significant RAO´s score-test value was retained, and the multivariable model was re-run. The parameters in the models were estimated using a gradient ascent machine learning algorithm, and the significance of these paremeters was assessed using the test of Wald. The goodness of fit of the models was established using the test of Hosmer and Lemeshow, which checks the significance of the difference between expected and observed values, so that non significant differences mean that the fit is good. We used IBM-SPSS Statistics 24 software package to perform the models and all the associated tests.

We subsequently updated the baseline disease model to explain the distribution of yellow fever cases in the early 21st century. Compared to the procedure described for the 20th-century model, we included a new block before the two ones mentioned above. Thus, the methodological sequence was as follows: (1) forcing the input, as a predictor variable, of the logit of the late 20th century baseline disease model (the logit being the linear combination of variables in the 20th-century model); (2) making a later stepwise selection of spatial and environmental variables; and (3) a stepwise addition of chorotypes that contribute to improving the model's likelihood. In this way, we took into account that the current spread of yellow fever is influenced by the inertia of previous situations. This is equivalent to assuming that there is temporal autocorrelation (i.e., disease cases in the early 21st century are more probable to occur in areas where they already occurred in the late 20th century). In the 21st-century model, the variables entering in blocks (2) and (3) represent the drivers potentially favouring the spread[34]. The preselection of variables for blocks (2) and (3) and the control for excessive multicollinearity between environmental variables were made as explained for the late 20th-century model.

**Vector models**. We produced a favourabuility model for each vector species for the late 20th century and for the early 21st century separately. We built multi-variable favourability models for urban vectors using the distribution of each urban mosquito species in the late 20th century and the spatial and environmental variables for the late 20th century, following the same procedure used for block (1) in the 20th-century baseline disease model. We also updated each urban vector model for the early 21st century as in the baseline disease model, using the procedure described for blocks (1) and (2).

A single model, referred to both the late 20th and the early 21st centuries, was made for sylvatic vectors, for the reasons explained above. Finally, we built up the vector models for the late 20th century and for the early 21st century by joining all individual vector models of each period using the fuzzy union[64] (i.e., considering for each hexagon the maximum value shown by any of the species models). This criterion was taken into account because, if the pathogen were present, the occurrence of a single vector species would involve potential for yellow fever transmission.

**Model fit assessment and validation of its predictive capacity**. Favourability models were assessed according to their classification and discrimination capacities respect to the training data set (i.e., to the observations used for model training). The classification capacity was based on two classification thresholds: $F = 0.5$, which represents the neutral favourability, and $F = 0.2$, below which the risk of transmission was considered to be low[61]. Six classification assessment indices were used[65]: (1) sensitivity (i.e., proportion of presences correctly classified in favourable hexagons), (2) specificity (i.e., proportion of absences correctly classified in unfavourable hexagons), (3) CCR (i.e., proportion of presences and absences correctly classified in favourable hexagons respectively), (4) TSS (that is sensitivity + specifity - 1), (5) underprediction rate (i.e., proportion of favourable areas that are recorded to have presences), and (6) overprediction rate (i.e., proportion of favourable areas that are not recorded to have presences). The discrimination capacity was assessed using the area under the receiver operating characteristic (ROC) curve (AUC)[66].

The validation of the predictive capacity of the late 20th century disease and transmission-risk models was done by evaluating, with the same indices used above, classification and discrimination capacities with respect to the cases of the period 2001−2020. The predictive capacity of the models for the early 21st century was validated with respect to the yellow fever cases reported in the period 2018−2020.

**Relative importance of the zoogeographical factor**. We estimated the pure contribution of non-human primates to the baseline disease model, i.e., how much of the variation in favourability for yellow fever cases was explained exclusively by the

zoogeographical factor, by performing a variation partitioning analysis[67]. This implied the use of the zoogeographic model and a spatio-environmental model constructed with the environmental and spatial variables that entered the baseline disease model. This approach also allowed us to calculate how much is the variation of the baseline disease model attributable simultaneously to the zoogeographical and other factors. We built maps identifying the zones where the non-human primates could increase yellow fever cases in humans, that is, where the presence of primates could favour the occurrence of yellow fever regardless of correlations with other factors. To map these areas we identified the hexagons that fulfilled these conditions: 1) favourability values for the baseline disease model were ≥ 0.2; and 2) the difference between the favourability values provided by the baseline disease model and the spatio-environmental model was positive and ≥ 0.01.

**Reporting summary**. Further information on research design is available in the Nature Research Reporting Summary linked to this article.

## Data availability

The sources for all data supporting the results of this study are cited in the main text, in Supplementary table 3, and in Supplementary data 1. The occurrence of yellow fever case reports in the spatial units employed here can be found in Supplementary data 2.

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

## Acknowledgements

We are grateful to Freya Shearer for sharing part of the georeferenced yellow fever cases considered in this research. This study was supported by the Project CGL2016-76747-R, of the Spanish Ministry of Economy, Industry and Competitiveness and the European Regional Development Fund, and by the Project B4-2021-14 (08.37.00.20.27) of the Research Plan of the University of Malaga. AA-S was supported by the FPU16/06710 grant of the Spanish Ministry of Education.

## Author contributions

A.A-S. and J.O. designed the research; A.A.-S., C.M-D., and J.O. made data search, curation and analysis; A.A-S., R.R., M.S. and J.O. interpreted the results; A.A-S, R.R, and J.O. wrote the paper; R.R., M.S. and J.O. revised the final manuscript.

## Competing interests

The authors declare no competing interests.
