## [Peer Review File · Communications Biology]

Reviewers' comments:

Reviewer #1 (Remarks to the Author):

In this manuscript the authors report their findings regarding the possible spread of yellow fever based upon the results of a spatial-temporal model that they developed, based upon previously published models. In simple terms, model is based upon a combination of habitat suitability for the vector, coupled with a disease model for the presence of yellow fever. The model uses fuzzy logic principles to develop a continuous variable – the favorability value – of the disease and vector models and uses a t-norm operation to resolve the intersection of the two favorability values for the final transmission model. The results and policy recommendations given by the authors appear to be well supported by their model outputs.

While the manuscript needs some copy-editing to improve clarity, my primary concern is that the overall description of the model needs to be improved to ensure that it is clear exactly how the vector and disease models are implemented. While having the source code available would be ideal, at a minimum the authors should include either pseudocode or a flowchart (perhaps both) to assist the reader in understanding the model. Much of this material could be delivered as part of the supplemental materials. Proper documentation of the model could also allow it to be modified for use in exploring other vector-borne pathogens.

- Major Points -

The authors description of their methodological approach and the algorithms they developed are unclear and can be difficult to parse in places. The manuscript would benefit greatly from the addition of pseudocode (or a flowchart) to clarify exactly how their model works. Some of this material could likely be included in supplemental materials to ensure that there is proper space to describe the processes.

The authors noted that they are using a Boolean to indicate the presence / distribution of reported cases of yellow fever in each hexagon (Lines 478 – 480). My concern is that it is unclear how this might skew the model – should a hexagon with one reported case of yellow fever be given the same weight as a hexagon with 1,000 cases?

It's unclear why the authors are using the divide between the 20th and 21st century as a crisp boundary since this also results in an apparently uneven amount of data available for the 20th century model (1970 – 2000, 30 years) and 21st century model (2001 – 2017, 16 years). The authors should expand upon this division of time as opposed approaches to the time division (e.g., 10-year increments, 20-year increments, etc.)

The section regarding the model assessment (Lines 203 – 254) is a bit difficult to parse and needs to be revised to improve the clarity. However, based upon my understanding of the author's assessment, it appears that their model is significantly biased towards over-prediction (i.e., identifying favorable locations as having yellow fever present), which is a point that should be explored more by the authors.

Since the author's R source code is central to the results of this manuscript, it should be made available for examination. At a minimum it appears that the authors are missing the required code availability statement.

It may be a limitation of the copy of the manuscript provide to reviewers, but it is very difficult to read some of the text that is present in the figures, and it can also be difficult to determine what is going on in the subfigures. This might also be a bit of a quibble, but I'm not sure using the set intersection symbol (Figures 2, 3) is entirely clear or appropriate since the authors are using fuzzy intersections (i.e., t-norm) and this could lead to confusion on the part of the reader, partially since this point is not clarified in the figure caption. The authors may wish to consider an alternative notation – or simply writing it out – to indicate that a fuzzy operation is being performed.

- Minor Points -

Lines 37 – 39 – You may want to give an example of some of the countries in Francophone Africa.

Lines 57 – 59 – “very complete” is vague, what about the cited work by Shearer et al. makes it very complete?

Lines 59 – 62 – This sentence might need to be rephrased since it doesn’t appear that Shearer et al. are arguing that the spread of the vector *A. aegypti* will lead to the spread of yellow fever, but rather that the presence of *A. aegypti* (intuitively) is one factor that contributes the receptivity of yellow fever transmission in new regions through the spread of the virus or through importation.

Lines 96 – 116 – The authors may want to considered restructuring most of the information in this paragraph to be presented as a table.

Lines 118 – 119 – “All the baseline ...” the way this sentence is phrased makes it unclear what the authors are trying to say.

Lines 125 – 127 – The point about yellow fever cases being associated with higher slopes in the late 20th century is interesting, is there any insight from the model or other literature in the field as to why this might be the case?

Lines 160 – 161 – Was any color coding applied to Figure 1 for the range of values between 0.2 and 0.5?

Lines 179 – 182 – This sentence is unclear and should be rephrased.

Lines 274 – 321 – This single paragraph need to be revised and broken up into two or three paragraphs to improve readability and clarity.

Lines 445 – 446 – Unclear exactly what the size of each hexagonal unit is – 7,774 km² each?

Lines 446 – 448 – Are you using a backend database such as PostgreSQL or MySQL? Or are you simply using the term ‘database’ to refer to the data structure that you are associating with each spatial location? Additionally, ‘instantiated’ would be a better to use than ‘prepared’ when discussing program execution.

Lines 448 – 449 – Presumably ArcMap 10.7?

Lines 457 – 475 - Since the authors are using additional data sets beyond what is in Shearer et al. it would be nice if there was supplemental table that included a complete list as well as the relevant citation and/or URL (DOI) of the data sources. Such as table could feasibility replace most of this block.

Lines 543 – 547 – The authors could be a bit more precise in their use of terminology since it appears they are referring to a minimum t-norm ($T_{\min}(a, b) = \min\{a, b\}$) for the intersection.

The overall phrasing of the sentence is also a bit awkward and should be revised for clarity.

Lines 555 – 557 – The authors need to elaborate more as to what variables from remote sensing data were used since Supplemental Table 2 is difficult to parse. I also note that the authors argue that these data points are only accessible for the 21st century; however, the authors should expand upon this point since Landsat imagery has been used going back to the 1980s for land use/landcover, landcover change studies at 30m resolution.

Lines 588 – 592 – “The rationale for this...” this sentence is a very awkwardly worded, but it appears the authors are stating that any spread of yellow fever indicated by the 20th century model is used as inputs to the 21st century model?

- Grammatical Issues -

Line 32 – “The Yellow fever...” inconsistent capitalization

Line 37 – “The yellow fever...” awkward phrasing

Lines 47 – 49 – “So, despite control policies...” awkward phrasing

Lines 62 – 64 – “So the spatial pattern...” awkward phrasing

Lines 140 – 143 – “However, this contribution...” quite a few typos in this sentence, also it is unclear what the authors mean by “undistinguishably explained”

Lines 446 – 448 – “The digital database...” awkward phrasing

Various spelling errors and typos throughout.

Reviewer #2 (Remarks to the Author):

The authors present an assessment of changes in the spatial distribution of yellow fever risk,

specifically noting the contributions of nonhuman primate species presence. A substantial spread in the area at risk of yellow fever occurrence is observed in both South America and Africa.

MAJOR COMMENTS

1. A brief summary of the methods is needed either at the end of the introduction section or the start of the results section. A reader reading the paper in the order it is written will need some overview of the methods used to be able to understand the results.

2. It is unclear what model was used to produce the results in Figure 1. The favorability score is presumably a function of the fitted logistic regression, but it is unclear which step of model fitting and which set of predictors was used. Is this model distinct from the others shown in Figure 2?

3. The rationale for separating the late 20th Century and early 21st Century appears to be based on the availability of remote sensing data, but the division at 2017/2018 needs justification. The early 21st Century data will include recent outbreaks in Angola, DRC, and Brazil. In Brazil, the recent outbreak occurred largely in locations that had not seen yellow fever for several years prior. Does this anomaly affect the results from this model?

4. What was the temporal resolution of the data analyzed? Was the outcome an indicator of yellow fever occurrence at any point within each time period or was each grid represented at multiple time points such as months or years? One related concern with using 2018-2020 data to validate the model fit using data from 2001-2017 is that the model was fit using data where grids had 17 years of time to acquire a yellow fever case, whereas the validation data set only contains 3 years.

MINOR COMMENTS

5. Lines 118-119: It is unclear what the expected and observed data are. If the authors are referring to the data used to fit the model and the validation data from the following time period, it would be better to keep the same naming convention.

6. Data availability is stated, but data used in analyses are not present. Particularly for the collection of yellow fever case data past 2016, it would be helpful to include, whether in the main text or the supplement, a list of sources/citations that contained the data collected. Additionally, the data availability lists two studies with data from 1970 to 2015, but only one is mentioned in the main text.

7. A data description is missing. What was the final sample size of the data analyzed? How many grids saw zero cases vs any cases?

8. The models are referred to as "multivariate" models in the text, but their descriptions seem as if they are actually multivariable models. See Hidalgo and Goodman (2013) for an explanation of this distinction.

9. Throughout, there is inconsistency in whether yellow fever is hyphenated.

Dear editor,

We have found in the reviews a very constructive feedback that has helped us to enhance the manuscript by correcting mistakes and improving explanations. The most relevant changes were:

- We have reinforced justifications for the time periods considered.
- We have included a flowchart of the methodological approach in the manuscript. In addition, we have included very detailed methodological explanations in the supplementary information to complement the flowchart.
- We have given details on methodological decisions that were incomplete or weakly justified.
- We have included a paragraph with a more detailed explanation of the meaning of the high overprediction of our models.
- We have improved figures following the reviewer's advices.
- We have corrected language mistakes and improved the English style in sentences that were difficult to understand.

Please, find below the comments of all reviewers arranged as we have found them in the revision letter, followed by our comments and responses which, just as this paragraph, were highlighted in blue. Line numbers we mention below make reference to the "cleaned" change-tracked manuscript, that is, numbers as they appear once the vertical lines on the left of the numbers is clicked.

Reviewers' comments:

Reviewer #1 (Remarks to the Author):

In this manuscript the authors report their findings regarding the possible spread of yellow fever based upon the results of a spatial-temporal model that they developed, based upon previously published models. In simple terms, model is based upon a combination of habitat suitability for the vector, coupled with a disease model for the presence of yellow fever. The model uses fuzzy logic principles to develop a continuous variable – the favorability value – of the disease and vector models and uses a t-norm operation to resolve the intersection of the two favorability values for the final transmission model. The results and policy recommendations given by the authors appear to be well supported by their model outputs.

While the manuscript needs some copy-editing to improve clarity, my primary concern is that the overall description of the model needs to be improved to ensure that it is clear exactly how the vector and disease models are implemented. While having the source code available would be ideal, at a minimum the authors should include either pseudocode or a flowchart (perhaps both) to assist the reader in understanding the model. Much of this material could be delivered as part of the supplemental materials. Proper documentation of the model could also allow it to be modified for use in exploring other vector-borne pathogens.

- Major Points –

The authors description of their methodological approach and the algorithms they developed are unclear and can be difficult to parse in places. The manuscript would benefit greatly from the addition of pseudocode (or a flowchart) to clarify exactly how their model works. Some of this material could likely be included in supplemental materials to ensure that there is proper space to describe the processes.

We agree. We have made a flowchart with the methodological framework for yellow fever transmission risk modelling (see Fig. 4.). In addition, we have included very detailed methodological explanations in Supplementary Methods that complements the flowchart.

The authors noted that they are using a Boolean to indicate the presence / distribution of reported cases of yellow fever in each hexagon (Lines 478 – 480). My concern is that it is unclear how this might skew the model – should a hexagon with one reported case of yellow fever be given the same weight as a hexagon with 1,000 cases?

With this modelling procedure, our aim was to calculate values of favourability for the occurrence of yellow fever transmission cases, disregarding of the number of cases was high or low. Secondly, studies based on species distribution modelling (VanDerWal *et al. Amer. Nat.* 174, 282-291 (2009)); Muñoz *et al. Divers. Distrib.* 21, 1388-1400 (2015)) have suggested that factors favouring the presence of the modelled item could also favour high abundances. So, high favourability values might point to high maximum abundance of cases, although we would need further analysis (which will be the task of future works) for testing this possibility.

Anyway, the occurrence of a unique case record at a hexagon could mean that the disease prevalence was low, but it could also mean that the sampling efforts in the area were low. Although we cannot avoid false absences throughout the hexagons grid (just as we recognize in the discussion, lines 294-312), at least we can avoid the noise caused by under-sampled areas within the hexagons. So, a major advantage of analysing presences using a grid approach was the possibility to avoid autocorrelations caused by sampling bias within these units.

It's unclear why the authors are using the divide between the 20th and 21st century as a crisp boundary since this also results in an apparently uneven amount of data available for the 20th century model (1970 – 2000, 30 years) and 21st century model (2001 – 2017, 16 years). The authors should expand upon this division of time as opposed approaches to the time division (e.g., 10-year increments, 20-year increments, etc.)

The reviewer is right to note that there is an uneven amount of data available for the 20th-century model and the 21st-century model. However, considering same-length time periods would not provide an even distribution of data. In fact, the information available for the last 30 years of the 20th century points to the occurrence of yellow fever cases in 218 hexagons, whereas the number of hexagons showing cases in the first 16 years of the 21st century was 493. This might partially reflect the current spread of the risk of disease in some regions, but it surely denotes as well that the availability of information is now higher than it was 40 years ago. We tried to simplify the question by splitting the time line in a single temporal point, which, although arbitrary, has been justified according to historical facts with relevance for the disease (as we explain in the new version of the manuscript, see lines 443-449: 1) Distributional changes in the ranges of the *Aedes aegypti* and *Aedes albopictus* vectors⁵¹ (Liu-Helmerson *et al. Front. Public Heal.* 7, (2019)(51)); 2) after 1999, the yellow fever genotype I has spread outside the endemic regions, and the genotype I modern-lineage has caused all major yellow fever outbreaks detected in non-endemic regions of South America since 2000 (Mir, D. *et al. Sci. Rep.* 7, 1-9 (2017)(13)); 3) the maximum potential of globalization was realised at the beginning of the 21st century with the opening of international borders, the widespread access to the Internet and to cell phones, and the generalization of online travel booking and of low-cost flights (Aliaga-Samanez, A. *et al. PloS Negl. Trop. Dis.* 15, e0009496 (2021) (34)).

The section regarding the model assessment (Lines 203 – 254) is a bit difficult to parse and needs to be revised to improve the clarity. However, based upon my understanding of the author's assessment, it appears that their model is significantly biased towards over-prediction (i.e., identifying favorable locations as having yellow fever present), which is a point that should be explored more by the authors.

We have rewritten the definitions of all the classification assessment indices in order to improve the clarity (see lines 629-635).

In order to deepen the discussion on the high over-prediction rates found in our models, we have modified completely the paragraph between lines 294 and 312: “The interpretation of models based on incomplete information must be made with caution. Sampling bias could lead to interpretation mistakes if the model output maximized the geographic fit between recorded disease cases and the resulting predictions. The combination of logistic regressions and the favourability function, used to get the outputs of our models, do not usually tend to overfitting (Olivero, J. *et al. Anim. Biodivers. Conserv.* 39, 99-114 (2016) (32)), hence these models showed high over-prediction rates (table 1), that is, a high proportion of favourable areas that are not recorded to have had cases. Part of this over-prediction might be explained by transmission risks forecasted in areas prone to be endemic in the short term, as shown by the fact that the over-prediction rate of the 20th-century model was around 0.9 (table 1), but it decreased until about 0.8 when this model's prediction capacity was tested according to the 21st-century cases (table 2). However, our model outputs are also consistent with the belief that the reporting of yellow fever cases could underestimate the actual number of cases, this being up to 500 times higher than reported in Africa, and around 10 times higher in South America (Shearer, F. M. *et al. Lancet Glob. Heal.* 6, e270-e278 (2018) (6); Barret, A. D. T. & Higgs, S. *Annu. Rev. Entomol.* 52, 209-229 (2007) (33)). Anyway, there are limits for any kind of model trying to analyse the geography of infection patterns for a virus showing high spatio-temporal dynamism, that is affected by the emergence of new lineages, by the changing distribution of *Aedes* mosquitoes (Aliaga-Samanez, A. *et al. PloS Negl. Trop. Dis.* 15, e0009496 (2021) (34)), as much as by revisions of vaccination strategies. So, our models have to be interpreted in the current historical context, as they were designed for this specific spatio-temporal window.”

Since the author's R source code is central to the results of this manuscript, it should be made available for examination. At a minimum it appears that the authors are missing the required code availability statement.

We used R for building zoogeographic variables based on chorotypes, using the RMacoqui package; and for building the worldwide grid of hexagons, using the dggridR package. The analyses performed by RMacoqui correspond to a method that was first published by Márquez *et al.* (1997), and was updated and contextualized under a fuzzy logic framework by Olivero *et al.* (2011), when this R package was developed. We mentioned the link: <http://rmacoqui.r-forge.r-project.org/> in the manuscript, which gives free access to the source code. Now, we have also added the script employed in the Supplementary Methods. RMacoqui has been used in papers such as Ferro *et al.* (2017), Olivero *et al.* (2017), Aliaga-Samanez *et al.* (2021), and Caballero-Herrera *et al.* (2021).

In relation to the dggridR package, it was published by Barnes and Sahr in 2017, and has been used in several articles such as Ferreira *et al.* (2020), Bousquin (2021), and Aliaga-Samanez *et al.* (2021). We have included in the manuscript the reference of this package and the link to it in the internet (<https://github.com/r-barnes/dggridR>) (line 433).

Cited references:

-Márquez A. L., Real R., Vargas J. M. & Salvo A. E. On identifying common distribution patterns and their causal factors: a probabilistic method applied to pteridophytes in the Iberian Peninsula. *J. Biogeogr.*, 24, 613-631 (1997).

-Olivero, J., Real, R. & Márquez, A. L. Fuzzy chorotypes as a conceptual tool to improve insight into biogeographic patterns. *Syst. Biol.* 60, 645-660 (2011).

-Ferro, I., Navarro-Sigüenza, A. G., & Morrone, J. J. Biogeographical transitions in the Sierra Madre Oriental, Mexico, shown by chorological and evolutionary biogeographical affinities of passerine birds (Aves: Passeriformes). *J. Biogeogr.*, 44, 2145-2160 (2017).

- Olivero, J., Fa, J. E., Real, R., Farfán, M. Á., Márquez, A. L., Vargas, J. M., ... & Nasi, R. Mammalian biogeography and the Ebola virus in Africa. *Mamm Rev.*, 47, 24-37 (2017).

- Aliaga-Samanez, A., Cobos-Mayo, M., Real, R., Segura, M., Romero, D., Fa, J. E., & Olivero, J. Worldwide dynamic biogeography of zoonotic and anthroponotic dengue. *PLoS Negl. Trop. Dis.* 15(6), e0009496 (2021)

- Caballero- Herrera, J. A., Olivero, J., von Cosel, R., & Gofas, S. An analytically derived delineation of the West African Coastal Province based on bivalves. *Divers. Distrib.* 00, 1-15 (2021)

-Barnes R. dggridR: Discrete Global Grids for R. R package version 2.0.4. Zenodo. 2017. <https://doi.org/10.5281/zenodo.1322866>

-Bousquin, J. Discrete Global Grid Systems as scalable geospatial frameworks for characterizing coastal environments. *Environ. Model. Softw.*, 146, 105210 (2021).

-Ferreira, L. N., Vega-Oliveros, D. A., Cotacallapa, M. et al. Spatiotemporal data analysis with chronological networks. *Nat Commun.* 11, 4036 (2020). <https://doi.org/10.1038/s41467-020-17634-2>

It may be a limitation of the copy of the manuscript provide to reviewers, but it is very difficult to read some of the text that is present in the figures, and it can also be difficult to determine what is going on in the subfigures. This might also be a bit of a quibble, but I'm not sure using the set intersection symbol (Figures 2, 3) is entirely clear or appropriate since the authors are using fuzzy intersections (i.e., t-norm) and this could lead to confusion on the part of the reader, partially since this point is not clarified in the figure caption. The authors may wish to consider an alternative notation – or simply writing it out – to indicate that a fuzzy operation is being performed.

We have increased the size of small texts in all the figures. We have also replaced “intersection” with “fuzzy intersection” in the figure captions (where we see that the intersection symbol has been mistakenly replaced with a rectangle). We agree that this will be clarifying, although we will keep the intersection symbol because it is appropriate in fuzzy logic as well (e.g., see Dubois & Prade. Operations in a fuzzy-valued logic. *Inf. Control.* 43, 224-240, 1979).

- Minor Points –

Lines 37 – 39 – You may want to give an example of some of the countries in Francophone Africa.
Done

Lines 57 – 59 – “very complete” is vague, what about the cited work by Shearer et al. makes it very complete?

We have eliminated “is a very complete pathogeographic analysis that”

Lines 59 – 62 – This sentence might need to be rephrased since it doesn't appear that Shearer et al. are arguing that the spread of the vector *A. aegypti* will lead to the spread of yellow fever, but rather that the presence of *A. aegypti* (intuitively) is one factor that contributes the receptivity of yellow fever transmission in new regions through the spread of the virus or through importation.

We have rephrased the sentence in order to improve clarity: “The study suggests that the presence of *A. aegypti*, combined with the zoonotic potential for infections (given the presence of potential primate hosts), contributes the receptivity of yellow fever transmission in new regions such as Asia through the spread of the virus or through importation.” (lines 59-62).

Lines 96 – 116 – The authors may want to considered restructuring most of the information in this paragraph to be presented as a table.

We have replaced the list of primate genera per chorotype in the main text with the supplementary table 1.

Lines 118 – 119 – “All the baseline ...” the way this sentence is phrased makes it unclear what the authors are trying to say.

We have rephrased the sentence in order to improve clarity: “All the baseline disease favourability models fitted the observed distribution of yellow fever cases, according to Hosmer & Lemeshow’s test of goodness of fit” (lines 104-106).

Lines 125 – 127 – The point about yellow fever cases being associated with higher slopes in the late 20th century is interesting, is there any insight from the model or other literature in the field as to why this might be the case?

We have looked for it, but have not found any reference to this relation in the literature.

Lines 160 – 161 – Was any color coding applied to Figure 1 for the range of values between 0.2 and 0.5?

There was a mistake in the legend of figure 1A. Yellowish green in the map represents “low-intermediate” favourability values between 0.2 and 0.5. These models based only on chorotypes did not show values lower than 0.2. The mistake in the legend has been corrected (lines 143-151).

Lines 179 – 182 – This sentence is unclear and should be rephrased.

We have rephrased this sentence as follows: “In the early 21st century, the risk of yellow fever transmission in South America could have increased in Paraguay, in some provinces of northern Argentina (e.g., Misiones and Corrientes), and in the “Mata Atlantica” in the south-east of Brazil (Fig. 2)” (lines 169-171).

Lines 274 – 321 – This single paragraph need to be revised and broken up into two or three paragraphs to improve readability and clarity.

We have broken up it into three paragraphs, and have deeply modified the texts between lines 262-271, 272-293 and 294-312.

Lines 445 -446 – Unclear exactly what the size of each hexagonal unit is – 7,774 km² each?

We have rephrased as follows: “We latticed the data (Cressie, N. A. C. Statistics for spatial data. (John Wiley & Sons, Inc., 1993) (49)) using a worldwide grid composed of 18,874 hexagonal 7,774-km² units” (lines 432-433).

Lines 446 – 448 – Are you using a backend database such as PostgreSQL or MySQL? Or are you simply using the term ‘database’ to refer to the data structure that you are associating with each spatial location? Additionally, ‘instantiated’ would be a better to use than ‘prepared’ when discussing program execution.

We have deleted the term “database” from this sentence, which has been rephrased completely: “All the information we processed on yellow fever cases, on urban and sylvatic vectors presences, and on zoogeographic, spatial and environmental variables (see details on this information below) was aggregated at this spatial resolution” (lines 433-436). “Database” was also mentioned in line 479, and has been now replaced with “dataset”.

Lines 448 – 449 – Presumably ArcMap 10.7?

Corrected

Lines 457 – 475 - Since the authors are using additional data sets beyond what is in Shearer et al. it would be nice if there was supplemental table that included a complete list as well as the relevant citation and/or URL (DOI) of the data sources. Such as table could feasibility replace most of this block.

Done. A new Supplementary Data 1 has been provided with the sources of all data on yellow fever cases we have collected in order to complement Shearer et al.’s (2018) data set.

Lines 543 – 547 – The authors could be a bit more precise in their use of terminology since it appears they are referring to a minimum t-norm ($T_{\min}(a, b) = \min\{a, b\}$) for the intersection. The overall phrasing of the sentence is also a bit awkward and should be revised for clarity.

We have rewritten completely the way values for risk of transmission are calculated using the fuzzy intersection, and have added a reference supporting it (lines 545-550): “The risk of transmission was assessed by combining a first model describing areas favourable to the presence of yellow fever, i.e., the “baseline disease model”; and another model describing areas favourable to the presence of mosquitoes known to act as vectors, i.e., the “vector model”. For this combination, we used the fuzzy intersection (Dubois & Prade. Operations in a fuzzy-valued logic. Inf. Control. 43, 224-240, 1979 (62)), i.e., the transmission risk at each hexagon was valued at the minimum between favourability in the baseline disease model and favourability in the vector model.”

Lines 555 – 557 – The authors need to elaborate more as to what variables from remote sensing data were used since Supplemental Table 2 is difficult to parse. I also note that the authors argue that these data points are only accessible for the 21st century; however, the authors should expand upon this point since Landsat imagery has been used going back to the 1980s for land use/landcover, landcover change studies at 30m resolution.

We have rewritten the sentence regarding to the variables only employed in the enhanced 21st-century models: “we included, in the variable set, predictors that are only accessible for the 21st century (e.g., high-resolution population density, livestock, irrigation, infrastructures, intact forest, and GlobCover land cover classes; see Supplementary Table 3)” (lines 558-561). Besides, mentioning Supplementary Table 2 instead of 3, as we did, was a mistake we have corrected. Suppl. Table 3 provides details on the source of all variables, and marks with either two or three asterisks those variables that were used only in the enhanced 21st-century models.

Lines 588 – 592 – “The rationale for this...” this sentence is a very awkwardly worded, but it appears the authors are stating that any spread of yellow fever indicated by the 20th century model is used as inputs to the 21st century model?

We have rewritten this paragraph completely in order to improve readability: “In this way, we took into account that the current spread of the yellow fever is influenced by the inertia of previous situations. This is equivalent to assuming that there is temporal autocorrelation (i.e., disease cases in the early 21st century are more probable to occur in areas where they already occurred in the late 20th century). In the 21st-century model, the variables entering in blocks (2) and (3) represent the drivers potentially favouring the spread (Aliaga-Samanez, A. *et al. PloS Negl. Trop. Dis.* 15, e0009496 (2021) (34))” (lines 600-605).

- Grammatical Issues –

Line 32 – “The Yellow fever...” inconsistent capitalization
Corrected throughout the text.

Line 37 – “The yellow fever...” awkward phrasing

We have modified the phrase, replacing “the yellow fever was controlled” with “the number of yellow fever cases decreased” (line 36).

Lines 47 – 49 – “So, despite control policies...” awkward phrasing

We have modified the sentence: “Consequently, despite the fact that control policies were able to virtually eliminate the yellow fever in wide areas of the globe, the WHO insists in stating that prevention efforts should not be abandoned” (lines 46-49).

Lines 62 – 64 – “So the spatial pattern...” awkward phrasing

We have modified the sentence: “So, if the current situation points to a yellow fever geographic spread, the forecasting of future trends will need pathogeographical analyses based on the spatio-temporal context.” (lines 62-64).

Lines 140 – 143 – “However, this contribution...” quite a few typos in this sentence, also it is unclear what the authors mean by “undistinguishably explained”

We have modified the phrase: “Nevertheless, chorotypes could explain up to 11.5%, because 11.1% of the variation in favourability can be as much attributed to the presence of primate chorotypes as to the spatial/environmental factor (Fig. 1b)” (lines 126-128).

Lines 446 – 448 – “The digital database...” awkward phrasing

Various spelling errors and typos throughout.

Throughout the text, we have either deleted “digital database” or replaced “database” with “dataset”.

Reviewer #2 (Remarks to the Author):

The authors present an assessment of changes in the spatial distribution of yellow fever risk, specifically noting the contributions of nonhuman primate species presence. A substantial spread in the area at risk of yellow fever occurrence is observed in both South America and Africa.

MAJOR COMMENTS

1. A brief summary of the methods is needed either at the end of the introduction section or the start of the results section. A reader reading the paper in the order it is written will need some overview of the methods used to be able to understand the results.

We agree with the reviewer. We have made a flowchart with the methodological steps of our approach to yellow fever transmission risk modelling (see Fig. 4 and Supplementary Methods).

2. It is unclear what model was used to produce the results in Figure 1. The favorability score is presumably a function of the fitted logistic regression, but it is unclear which step of model fitting and which set of predictors was used. Is this model distinct from the others shown in Figure 2?

We agree that the meaning of Figure 1 was confusing. First, there was a big mistake in the description of Fig. 1 **a**; and second, keys for identifying differences between **a** and **b** were needed for clarity. We have rewritten completely the legend in this way: “**a** Model of favourability for the occurrence of yellow fever cases according to the presence of non-human primate chorotypes (i.e. zoogeographic model) [the scale for favourability values is: high ($F > 0,8$); high-intermediate ($0,5 \leq F \leq 0,8$); low-intermediate ($0,2 \leq F < 0,5$)]. **b** Partial contribution of primates on the presence of yellow fever cases in humans [the numbers are percentages of contribution to the distribution of favourability in the disease models (Z: zoogeographic factor, S/E: spatial/environmental factor)]. The maps in **a** represent the areas where the primate presence could favour the occurrence of disease cases in humans, although correlations with other factors influencing the primate biogeography (such as climate, topography or land cover) might be involved in this relation. Instead, the maps in **b** highlight the areas where the presence of primates could favour the occurrence of yellow fever regardless of correlations with other factors.” We have also included a clarification for this in the methods section (see lines 143-151).

3. The rationale for separating the late 20th Century and early 21st Century appears to be based on the

availability of remote sensing data, but the division at 2017/2018 needs justification. The early 21st Century data will include recent outbreaks in Angola, DRC, and Brazil. In Brazil, the recent outbreak occurred largely in locations that had not seen yellow fever for several years prior. Does this anomaly affect the results from this model?

A main objective of this research was to get models able to predict future trends given that the geographic distribution of the yellow fever transmission risk is changing, hence the anomalies in central Africa and Brazil. The interesting thing is that the late 20th century model already predicted the risk of yellow fever transmission in the eastern coasts of Brazil, despite cases in these areas had not been recorded yet; and the 20th century model also predicted the presence of risk in DRC, at least in the north, before cases were reported in that country. So, the 21st-century anomalies could represent a geographic trend that could have been predicted before it happened, hence the utility of modelling. Nevertheless, we decided to validate the predictive capacity of the 21st model as well, for which some final records had to be retained for the test. The training data set for an updated model for the 21st century should include yellow fever observations in the newly endemic areas, and this implied considering some years later than 2014. Our decision of choosing the 2017/2018 limit could seem arbitrary, but it gave us three years with yellow fever case records in south-west Brazil (two in Angola and DRC) in the training data set, and three later years for predictive testing purposes. We have explained this in the lines 449-453 of the new version.

4. What was the temporal resolution of the data analyzed? Was the outcome an indicator of yellow fever occurrence at any point within each time period or was each grid represented at multiple time points such as months or years? One related concern with using 2018-2020 data to validate the model fit using data from 2001-2017 is that the model was fit using data where grids had 17 years of time to acquire a yellow fever case, whereas the validation data set only contains 3 years.

To the first question, the spatial resolution was 1970-2000 for the 20th century model and 2001-2017 for the 21st century model. So, the local output of a transmission-risk model is referred to any time within the model training period. Nevertheless, as was said in the answer to the previous question, our models showed to have predictive capacity. So, outputs pointing to high risk in areas where cases have not been recorded might suggest the proximity of future changes. We hope that the new methodological details we are giving in this version may help (see Fig. 4.)

About the second reviewer's comment, we have not validated the model fit using 2018-2020 data. Instead, the 21st century model fit was assessed using 2001-2017 data (i.e., the set of yellow fever cases used for model training); whereas 2018-2020 data were used for validating the model predictive capacity (see answer to the previous question). In order to improve clarity, we have modified the title of one of the method's subsections: "Model fit assessment and validation of its predictive capacity" (line 623); and also its starting sentence: "Favourability models were assessed according to their classification and discrimination capacities respect to the training data set (i.e., to the observations used for model training)" (lines 624-626).

MINOR COMMENTS

5. Lines 118-119: It is unclear what the expected and observed data are. If the authors are referring to the data used to fit the model and the validation data from the following time period, it would be better to keep the same naming convention.

We have rewritten this sentence completely. So, "All the baseline disease models fitted the observed data, as there was no significant difference between the expected and the observed data" has been replaced with "All the baseline disease favourability models fitted the observed distribution of yellow fever cases, according to Hosmer & Lemeshow's test of goodness of fit" (lines 104-106). Now that we have explained better, in the methods section, the difference between model fit assessment and validation of the predictive capacity (see answers to previous questions), we think this sentence could be interpreted correctly.

6. Data availability is stated, but data used in analyses are not present. Particularly for the collection of yellow fever case data past 2016, it would be helpful to include, whether in the main text or the supplement, a list of sources/citations that contained the data collected. Additionally, the data availability lists two studies with data from 1970 to 2015, but only one is mentioned in the main text.

Two new tables, Supplementary Data 1 and Supplementary Data 2, have been provided. The former contains the sources and links for all data on yellow fever cases we have collected in order to complement Shearer et al.'s (2018) data set; the latter contains the centroid-coordinates of all hexagons known to have reported yellow-fever cases (i.e., the “presence” data set considered here for disease cases).

7. A data description is missing. What was the final sample size of the data analyzed? How many grids saw zero cases vs any cases?

In total, we used 18,874 hexagonal 7,774-km² units (line 432). For the late 20th century, we had 218 presence-hexagons and 18,656 absence-hexagons; for the early 21st century, we had 493 presence-hexagons and 18,381 absence-hexagons. This information is now available in lines 478-482.

8. The models are referred to as “multivariate” models in the text, but their descriptions seem as if they are actually multivariable models. See Hidalgo and Goodman (2013) for an explanation of this distinction.

Thank you very much. We have replaced “multivariate” with “multivariable” (lines 573, 584 and 610).

9. Throughout, there is inconsistency in whether yellow fever is hyphenated.

Thank you very much. We have eliminated hyphenation in yellow fever throughout the text.

REVIEWERS' COMMENTS:

Reviewer #2 (Remarks to the Author):

I commend the authors on addressing all of the concerns that I raised with the original draft, and my only remaining comment is a bit of a minor one, namely that there are still some grammatical issues throughout that need to be addressed. The most common appears to be the use of "the" before yellow fever. When referring to the disease the "the" is not needed; however, if the authors are referring to the yellow fever virus, then it should be written out in full (e.g., Lines 32 – 33 should read "The yellow fever virus is native to Africa...").

Reviewer #3 (Remarks to the Author):

The revised version of this manuscript shows greatly improved clarity in presenting the methods used and, by extension, the applicability and interpretability of the results. Overall, it reads well and presents very interesting research. A few minor points remain to improve pieces that are somewhat unclear or inconsistent.

-Lines 126-127 mention a westward expansion in Africa, but then list eastern African countries. Lines 279-281 then mention spread in Africa from the west to the east.

- Lines 202-203: TSS and CCR should be defined before their first use in the text.

REVIEWERS' COMMENTS:

Reviewer #2 (Remarks to the Author):

I commend the authors on addressing all of the concerns that I raised with the original draft, and my only remaining comment is a bit of a minor one, namely that there are still some grammatical issues throughout that need to be addressed. The most common appears to be the use of “the” before yellow fever. When referring to the disease the “the” is not needed; however, if the authors are referring to the yellow fever virus, then it should be written out in full (e.g., Lines 32 – 33 should read “The yellow fever virus is native to Africa...”).

Thank you so much. We have corrected grammatical errors. We have eliminated “the” from the wrong sites.

Reviewer #3 (Remarks to the Author):

The revised version of this manuscript shows greatly improved clarity in presenting the methods used and, by extension, the applicability and interpretability of the results. Overall, it reads well and presents very interesting research. A few minor points remain to improve pieces that are somewhat unclear or inconsistent.

-Lines 126-127 mention a westward expansion in Africa, but then list eastern African countries. Lines 279-281 then mention spread in Africa from the west to the east.

- Lines 202-203: TSS and CCR should be defined before their first use in the text.

Thank you so much. We have modified the wrong phrase by replacing “westward” with “eastward”. We have also included the complete definition of TSS and CCR as indicated.